# A Novel Dye-Sensitized Solar Cell Structure Based on Metal Photoanode without FTO/ITO

**DOI:** 10.3390/mi13010122

**Published:** 2022-01-13

**Authors:** Jianjun Yang, Xiaobao Yu, Yaxin Li, Guilin Cheng, Zichuan Yi, Zhi Zhang, Feng Chi, Liming Liu

**Affiliations:** 1College of Electron and Information, University of Electronic Science and Technology of China, Zhongshan Institute, Zhongshan 528402, China; yizichuan@163.com (Z.Y.); zz001@zsc.edu.cn (Z.Z.); chifeng@semi.ac.cn (F.C.); liulmxps@126.com (L.L.); 2School of Optoelectronic Science and Engineering, University of Electronic Science and Technology of China, Chengdu 610054, China; yuxiaobao16@126.com (X.Y.); leiyafangzhou@163.com (Y.L.); m13018210687_1@163.com (G.C.)

**Keywords:** DSSC, stainless steel, titanium, titanium dioxide, metal photoanode

## Abstract

Traditional dye-sensitized solar cells (DSSC) use FTO/ITO containing expensive rare elements as electrodes, which are difficult to meet the requirements of flexibility. A new type of flexible DSSC structure with all-metal electrodes without rare elements is proposed in this paper. Firstly, a light-receiving layer was prepared outside the metal photoanode with small holes to realize the continuous oxidation-reduction reaction in the electrolyte; Secondly, the processing technology of the porous titanium dioxide (TiO_2_) film was analyzed. By testing the J–V characteristics, it was found that the performance is better when the heating rate is slow. Finally, the effects of different electrode material combinations were compared through experiments. Our results imply that in the case of all stainless-steel electrodes, the open-circuit voltage can reach 0.73 V, and in the case of a titanium photoanode, the photoelectric conversion efficiency can reach 3.86%.

## 1. Introduction

In recent years, metal and metal compound electrodes have been extensively studied in DSSC. However, due to their opacity, most of them are limited to the use of a single metal electrode [1,2,3,4,5,6]. Electrode cost is a problem in the research of photoanodes and counter electrodes (CE) [7,8]. The typical metal mesh electrode technology has not yet been industrialized, and its insufficient light-receiving layer area and complex process of TiO_2_ film are both limiting factors [9,10,11].

The photoanode in DSSC should have the functions to adsorb dyes [12] and transmit and collect photogenerated electrons. A good photoanode should have a large specific surface area and a nanoporous structure, which can absorb dyes to the maximum and transmit photogenerated electrons quickly, and have sufficient porosity to penetrate the electrolyte [13,14,15]. Because TiO_2_ is nontoxic, cheap, and has strong ability to adsorb dyes [16,17] and transfer electrons [18], it is widely used in the fabrication of photoanodes [19,20,21,22,23,24,25]. The introduction of special nanoparticles into a TiO_2_ matrix can improve the performance of DSSCs, such as adding nanodiamonds (NDs) and CdTe nanoparticles [26,27], but the scarcity of materials will increase production costs. The importance of CE lies in its high catalytic activity [28], such as the catalytic reduction of $I_{3}^{-}$ [29]. By using Pt to reduce the oxidation-reduction potential of the $I_{3}^{-}$/$I_{1}^{-}$ ion pair, it helps to increase the open-circuit voltage [30,31,32,33]. Compared with CE based on Pt, the optimized NDs layer has 78.52% equivalent performance [34]. The traditional sandwich DSSC uses expensive FTO electrodes, which reduce the cost performance, and at the same time, due to its rigidity, the battery loses its flexibility. As for the back-illuminated structure [35], light needs to pass through the CE and electrolyte before reaching the photoanode, and its efficiency is naturally limited [36]. The metal mesh used as CE can replace the transparent conductive film, but the incident light loss caused by the mesh cannot be ignored [37].

In this paper, cheap metal materials (stainless steel and titanium) are used to replace expensive rare-element electrodes to produce photoanode and CE substrates. The photoanode is prepared on the outside of the stainless steel/titanium substrate to enhance light absorption. The electrolyte passes through the photoanode with the light-receiving layer through specially designed small holes on the substrate. The experimental results show that the all-metal electrode flexible DSSC has almost the same open-circuit voltage as the traditional DSSC with FTO. The schematic diagram and physical drawing are shown in Figure 1.

## 2. Experiments

### 2.1. Experimental Materials

The 304 stainless-steel foil was customized by Hengrui company, and Titanium foil was customized by Yiyuan. Dust-free paper, PET, dust-free cloth, tweezers, glass plate, glass rod, scissors, hot-melt glue gun, aluminum foil paper, fresh-keeping film, 3M Scotch tape, scraper, screen printing, and other tools were also used. In addition, the drugs used were as follows: AE (absolute ethanol, >99%), TiO_2_ (>99%), Pt (>99%), I_2_ (>99%), LiI (>99%), PMII (>99%), TBP (>99%), GuSCN (>99%), N719 (>99%).

### 2.2. Preparation of Photoanode

First, holes were punched in the stainless-steel foil, according to a certain density and size, then cleaned up and placed on a glass plate to dry. The scotch tape was glued to the dry foil, leaving a 5 × 5 mm^2^ square surface which included four holes. TiO_2_ slurry was scraped on the foil carefully with a glass rod. The foil sample was sent into a muffle furnace for heating and taken out after natural cooling. The TiO_2_ was dyed with 0.4 mmol/L N719 solution dissolved in ethanol. After 24 h, the foil with TiO_2_ was taken out and cleaned with AE, and then air-dried. The flowchart is shown in Figure 2. The preparation steps of titanium photoanode and non-drilling FTO photoanode are the same as above.

### 2.3. Preparation of Counter Electrode

The clean stainless-steel foil was placed and fixed on a screen-printing table with a 200-mesh screen. Then it was scraped with chloroplatinic acid slurry. After a few minutes, the foil with chloroplatinic acid film was placed in a heating boat and put into a tubular annealing furnace. The temperature was increased to 400 °C for 30 min. The CE could then be used after natural cooling. The preparation of FTO CE is the same as the above steps.

### 2.4. Photosensitizer and Electrolyte Preparation

(1) Photosensitizer preparation: each 100 mL of photosensitizer contained 47.5 mg dye N719 dissolved in AE. After the proportioning, the solution was poured into a beaker sealed with plastic wrap film and equipped with a magnetic rotor. The solution was stirred with a magnetic stirrer for 30 min.

(2) Electrolyte preparation: each 100 mL of electrolyte contained I_2_ 1.523 g, LiI 8.031 g, GuSCN 0.708 g, TBP 6.761 g, and PMII 7.563 g. All of them were dissolved by acetonitrile, then the acetonitrile solution was poured into a clean beaker to stir for 20 min with a magnetic stirrer [10,38].

### 2.5. Device Assembly

First, the PET was cut into 3 × 3 cm^2^, cleaned with AE, and placed on a glass plate. A CE was placed on PET in the lower left corner with the platinum side facing up. Then, the CE was covered completely with a piece of 1.8 × 1.8 cm^2^ dust-free paper. Finally, the photoanode and another piece of PET were placed on top of them one after the other. The battery was sealed with a hot-melt glue gun. A syringe was used to insert the electrolyte through the small hole reserved on the PET, and the hole was sealed at the end.

### 2.6. Characterization

The morphology of TiO_2_ on stainless steel and titanium sheet was observed with a positive metallographic microscope (BX51M, OLYMPUS, Tokyo, Japan). The two-dimensional and three-dimensional morphology of TiO_2_ surface were obtained by an atomic force microscope (DSM14049BF-1, Bruker, Billerica, MA, USA). The 100× and 500× magnified images of TiO_2_ films were obtained by scanning electron microscope SEM (TESCAN, Brno, Czech Republic). The J-V curve of DSSC was obtained through a combination of OAI solar simulator (LCSS150, Zolix, Beijing, China) and KEITHLEY2400 digital source meter (Oriel 94023A, Newport). The external quantum efficiency of DSSC was obtained through the quantum efficiency test system (PEC-S20, Gifu, Japan). The UV-VIS spectrum of DSSC was obtained by UV-VIS Spectrometer (UV-24500, Avantes, Appeldom, The Netherlands).

## 3. Results and Discussion

First, the experiments confirmed that the design goal of an all-metal electrode without FTO/ITO can be realized via this new structure, which has an external light-receiving layer and small holes on the photoanode. Moreover, the application of PET and metal foils also ensure the flexibility of DSSC. The measurement data of the samples show that in order to obtain sufficient output, the combination of electrode materials and the preparation process of the TiO_2_ layer must be handled carefully.

### 3.1. Design of Holes on Photoanode

The small holes reserved on the photoanode are an important structure to ensure the penetration of electrolyte and the continued progress of the redox reaction. When the aperture is too large, it is obvious that the area coated with TiO_2_ on the photoanode will decrease, resulting in the loss of the light-receiving layer. Figure 3 shows the dimensions of the two verified holes.

During the scraping of TiO_2_, Scotch tape was used to surround a 5 mm × 5 mm area covering four small holes on the electrode. For the first type shown in Figure 3a, the area of the light-receiving layer is 17.93 mm^2^, accounting for 72% of the whole area of 25 mm^2^; for the other type shown in Figure 3b, that is 22.99 mm^2^, accounting for 92% of the entire area. The area ratio of these two types is 0.78:1.

The stainless-steel photoanodes and titanium photoanodes with these different apertures have been assembled in DSSC, respectively. By comparing the J–V characteristic curves of the four samples, it is found that the hole size has no influence on the open-circuit voltage. On the other hand, whether it is titanium or stainless steel, the current density of the battery with ϕ1.5 mm holes is about 81% of that with ϕ0.8 mm holes. Looking into the light-receiving area ratio 0.78:1, it is obvious that the two ratios are very close. Consequently, we can judge that the diffusion of ions in electrolyte does not require a large pore size. Therefore, the density and size of these holes should be limited as much as possible to obtain a larger light-receiving layer area.

### 3.2. Influence of Heating Rate

A muffle furnace with a controllable temperature curve was used, and two groups of electrodes with different heating rates were set for experiments. After scraping TiO_2_ paste, the stainless-steel electrode of group A rose to 500 °C from room temperature within 45 min, and the temperature of group B rose within 30 min. After natural cooling, the electrode was taken out, and the TiO_2_ was transferred to the metallographic microscope. Figure 4a,c shows these patterns of the group A electrode with 100 times and 500 times magnification, respectively. Figure 4b,d shows those of group B. It can be intuitively observed that when the temperature is raised faster, a large number of cracks and defects appear on the surface of the group B electrode, and the intact film area accounts for only about 60%. Compared with group A, it can be observed that when the heating rate is slow, the surface of TiO_2_ can be relatively flat, and the texture can be consistent with that of stainless steel.

Two sets of electrodes were used to assemble DSSC, respectively. The solar energy test system was used under AM1.5 standard sunlight. It can be seen from Figure 5 that when the heating rate of the prepared electrode is 16 °C/min (Group B), the short-circuit current density is only half of the case of 11 °C/min (Group A). According to the morphology of the electrode surface, it can be inferred that the large-area damage caused by the heating rate to the film cannot be ignored, which will damage the photoelectric conversion ability.

### 3.3. Comparison between Metal Photoanode Battery and FTO Electrode Batteries

Four groups of control experiments were set, as shown in Table 1. The experimental procedure was the same as those described in Section 2, only the different electrodes were assembled into four groups of batteries. It is worth noting that the photoanode light-receiving layer of the stainless-steel substrate faced the outside of the battery; on the contrary, that of the FTO substrate faced the inside.

The J–V characteristics of the four groups are shown in Figure 6a, and the performance indexes are shown in Table 2. The results imply that the difference of photoanode substrate is the main factor that causes the photoelectric conversion ability of the battery. The difference in the short-circuit current density is the most obvious. The short-circuit current densities of group A(SS) and B(SF) were 3.8 mA/cm^2^ and 5.2 mA/cm^2^, respectively. FTO was used as the photoanode substrate in group C(FS) and D(FF), and the short-circuit current density reached 17.0 mA/cm^2^ and 18.2 mA/cm^2^, respectively. The difference is nearly four-fold. The reasons for the huge difference should be diverse. First, the most likely problem is the interface. The interface between the metal photoanode and TiO_2_ may be not well-combined, and the work function may not match very well. Secondly, the quality of the TiO2 film may be not good enough. Thirdly, there is leakage. Some pretreatment on the surface of the metal substrate may further improve the conversion efficiency.

At the same time, it can be found that there is almost no influence on the change of the substrate of the CE, though it is undeniable that FTO still has a slight advantage in the current characteristics as a CE.

Another key indicator of the battery, the open-circuit voltage, can also be seen in Figure 6a and Table 2. These open-circuit voltages of the four groups of batteries A, B, C, and D are 0.73 V, 0.73 V, 0.76 V and 0.76 V, respectively. They are very similar.

In addition, we have further studied the reasons for the large differences in the short-circuit current density of the four groups. Figure 7a,c shows the morphology of TiO_2_ sintered on FTO, and Figure 7b,d shows that of the stainless-steel electrode under a scanning electron microscope. Firstly, it can be found from Figure 7b that using the same method as FTO, the porous titanium dioxide layer was successfully prepared on the stainless-steel electrode; however, comparing Figure 7a,b, the coverage area of the porous titanium dioxide layer prepared on FTO is denser. In Figure 7c, the TiO_2_ layer uniformly covers the FTO surface, and the porous structure is relatively uniform, but there are still some irregularities, which may be caused by the uneven particle size of P25-TiO_2_. In Figure 7d, the irregularity of the titanium dioxide layer on the stainless steel is clear and the caves are deeper.

Figure 8 shows the AFM characterization of the TiO_2_ layer on FTO and stainless steel, respectively. It can be observed that porous TiO_2_ on FTO has more details and is intensive, while TiO_2_ on stainless steel is smoother, except for some burrs. The average roughness Ra of the film on FTO is 0.0182 µm, and the average roughness Ra of the film on stainless steel is 0.0403 µm. The height difference of the stainless-steel surface is larger. It can also be seen from Figure 8d that the TiO_2_ layer on the stainless-steel surface has a large wave shape. Combined with the metallographic microscope diagram, it implies that the fluctuation of the film surface is caused by the texture of the metal surface, which is also the reason for its large maximum roughness.

### 3.4. Influence of Different Metal Electrodes

Titanium foil is the most suitable metal electrode material for DSSC, with the exception of stainless steel. The difference of the J–V curve between the titanium photoanode dye-sensitized battery and the stainless-steel photoanode dye-sensitized battery is shown in Figure 6b. In these batteries, stainless-steel electrodes prepared with platinum black are used as the CE.

It can be seen that the titanium electrode adopting this new structure has remarkable photoelectric conversion ability. In the case of the titanium photoanode, the open-circuit voltage decreases slightly, but the short-circuit current density and filling factor have been significantly improved. The short-circuit current density can reach 8.87 mA/cm^2^ which is 2.3 times that of stainless steel.

## 4. Conclusions

In order to eliminate the restriction of FTO/ITO in DSSC production, a new DSSC structure with all-metal electrodes and electrode preparation process is studied in this paper. First, our research shows that if the electrolyte has sufficient fluidity, the density and size of the holes should be restricted as much as possible to obtain a larger light-receiving layer area. Second, the heating rate for preparing TiO_2_ film should not be too fast. The short-circuit current density at a heating rate of 11 °C/min is twice that of a heating rate of 16 °C/min. Third, the metal electrodes have little influence on the open-circuit voltage of DSSC. In the case of a totally stainless-steel electrode, the open-circuit voltage can be 0.73 V, which is almost same as a traditional batterie. Moreover, the photoanode substrate material and the morphology of TiO_2_ film are key factors of short-circuit current density. When the titanium is used as a photoanode substrate, and there is no need for a complicated process and any additional treatment, the photoelectric conversion efficiency can reach 3.86%. This is 4.65 times higher than stainless steel. At the same time, the short-circuit current density can reach 8.87 mA/cm^2^, and the filling factor can reach nearly 57%. In summary, it can be concluded that an all-metal electrode structure with an external light-receiving layer and small holes on the photoanode can be used for DSSC without any FTO/ITO.

## Figures and Tables

**Figure 1 micromachines-13-00122-f001:**
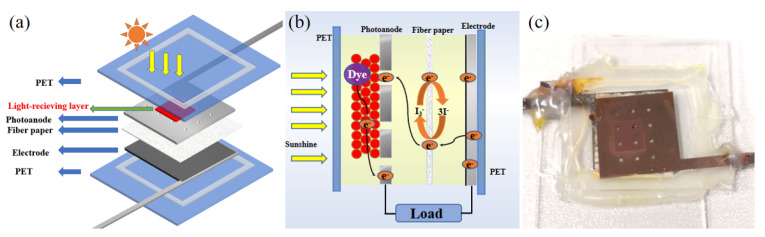
(**a**) Schematic diagram. A light-receiving layer is prepared on the outside of a metal electrode with small holes to form a photoanode; Fiber paper serves as an insulator and an electrolyte container; a platinum black film is prepared on a metal electrode to form the counter electrode; the battery is packaged with PET and injected with electrolyte. (**b**) Photoelectric conversion diagram. (**c**) Physical drawing.

**Figure 2 micromachines-13-00122-f002:**
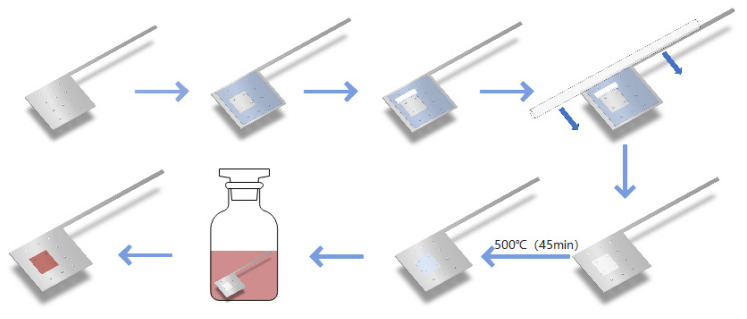
Photoanode preparation process.

**Figure 3 micromachines-13-00122-f003:**
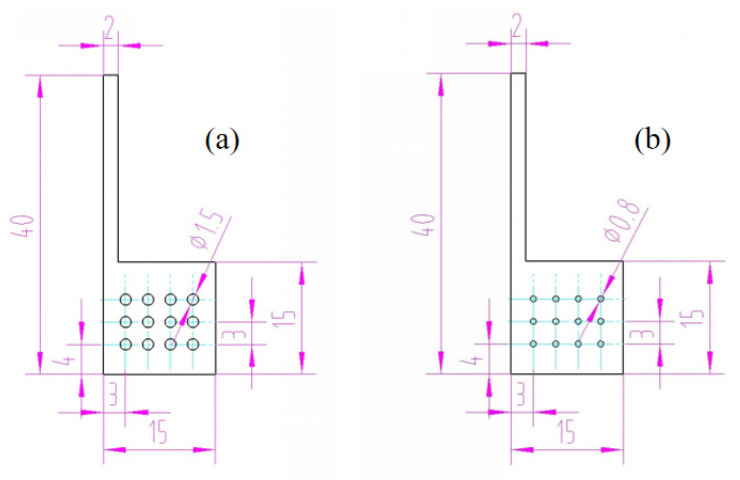
Drawing of holes on the photoanode. (**a**) holes in ϕ1.5 mm; (**b**) holes in ϕ0.8 mm. The units of the figures are in millimeters.

**Figure 4 micromachines-13-00122-f004:**
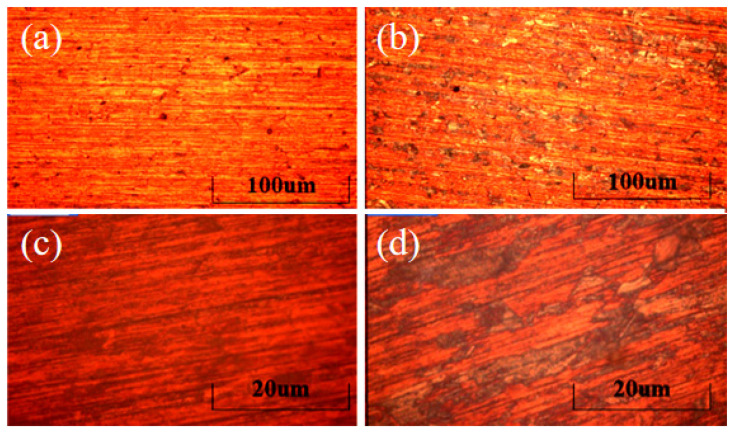
The morphology of TiO_2_ film through a BX51M microscope. (**a**) Group A, the heating rate is 11 °C/min and lasts 45 min, 100 times magnification. (**b**) Group B, the heating rate is 16 °C/min and lasts 30 min, 100 times magnification. (**c**) Group A with 500 times magnification. (**d**) Group B with 500 times magnification.

**Figure 5 micromachines-13-00122-f005:**
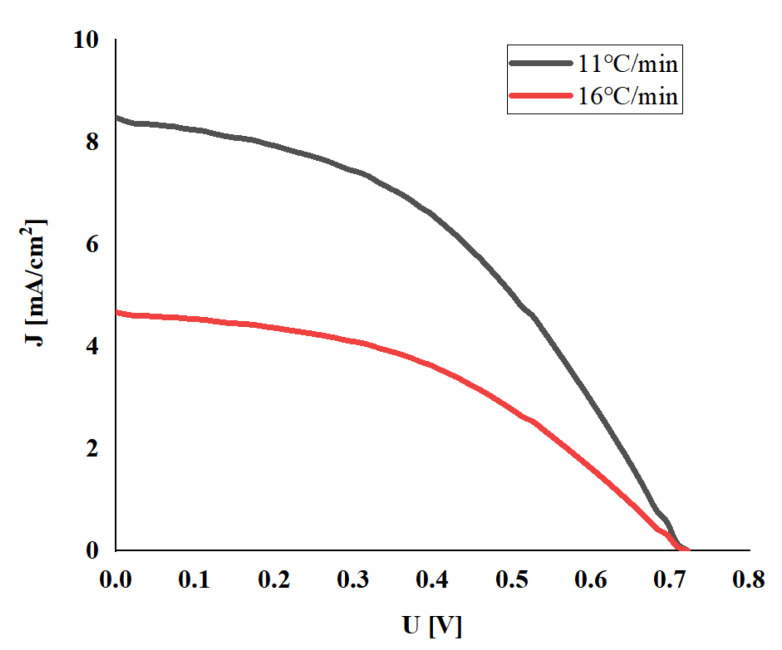
J–V Characteristics of DSSC with TiO_2_ film prepared at different heating rates.

**Figure 6 micromachines-13-00122-f006:**
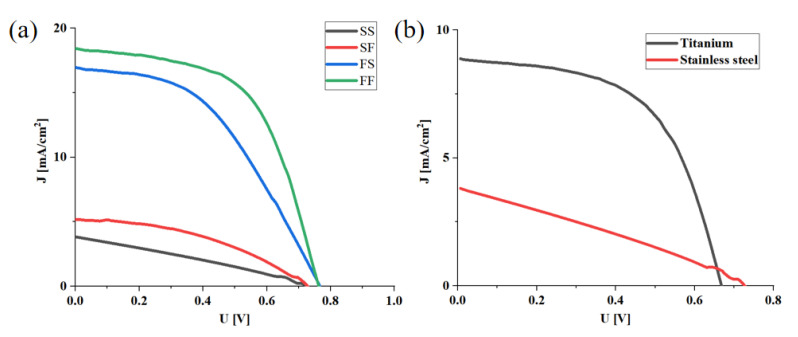
(**a**) J–V Characteristics curve of cells assembled with four different electrode combinations, (**b**) J–V characteristics curve of stainless-steel counter electrode cells assembled with two different metal photoanodes.

**Figure 7 micromachines-13-00122-f007:**
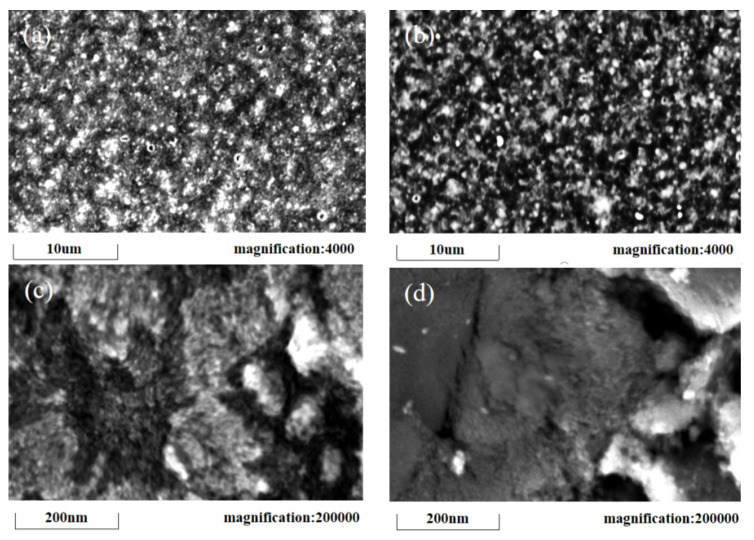
SEM of TiO_2_ films on different electrode surfaces: (**a**) FTO surface, 4000 times magnification; (**b**) stainless steel surface, 4000 times magnification; (**c**) FTO surface, 200,000 times magnification; (**d**) stainless steel surface, 200,000 times magnification.

**Figure 8 micromachines-13-00122-f008:**
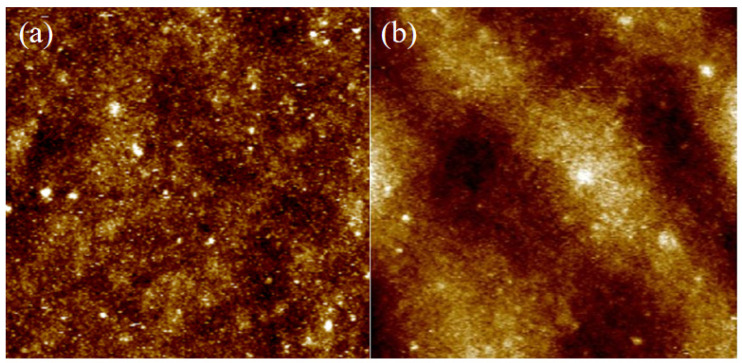
AFM of TiO_2_ thin film on different electrode surfaces: (**a**) TiO_2_ film morphology on FTO; (**b**) TiO_2_ film morphology on stainless steel; (**c**) AFM 3D diagram on FTO; (**d**) AFM 3D diagram on stainless steel.

**Table 1 micromachines-13-00122-t001:** Four experiment groups with different electrode combination.

Experiment Group	Photoanode Base Material	Counter Electrode Base Material
A (Steel/Steel, SS)	Stainless steel	Stainless steel
B (Steel/FTO, SF)	Stainless steel	FTO
C (FTO/Steel, FS)	FTO	Stainless steel
D (FTO/FTO, FF)	FTO	FTO

**Table 2 micromachines-13-00122-t002:** Performance of different electrode combination batteries.

Electrode Combination	Jsc (mA/cm^2^)	Voc (V)	FF (%)	η (%)
A (Steel/Steel)	3.8	0.73	30.1	0.83
B (Steel/FTO)	5.2	0.73	35.0	1.32
C (FTO/Steel)	17.0	0.76	55.0	7.11
D (FTO/FTO)	18.2	0.76	59.7	8.26

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
