# Peer review of "A Novel Dye-Sensitized Solar Cell Structure Based on Metal Photoanode without FTO/ITO"

_micromachines, 2022, doi:10.3390/mi13010122_

Round 1
Reviewer 1 Report
The manuscript "A Novel Dye Sensitized Solar Cell Structure Based on Metal
Photoanode without FTO/ITO" should highlight the significance and/or shortcomings of the new design presented in it. What are the reasons of poor performance of the DSSCs with metal based photoanodes?
The manuscript should be focused on proving the significance of new proposed design (if any).
The grammatical and linguistic errors should be removed.
Reviewer 2 Report
the concept of using metals as substrate has been presented nicely and research design has been properly elaborated. The introduction section may be improved using some recent research papers. some of the relevent references are as under
- https://www.sciencedirect.com/science/article/pii/S0038092X19312861
- https://link.springer.com/article/10.1007/s13369-017-2979-z
- https://www.sciencedirect.com/science/article/pii/S1386947717314558
furthermore, as the electron transfer has been discussed properly throughout the text. I would suggest to study the electrochemical impedance of the as prepared DSSCs. this will provide detailed characteristics of electron tramsfer
Reviewer 3 Report
In this article, the authors have employed metal foil (stainless steel, titanium) as conducting substrates instead of FTO/ITO for the fabrication of DSSCs. The study seems to be insufficient and there is no significant results are obtained. The authors showed (Table-2) that when both the substrates are FTO the short circuit current density and power conversion efficiency are high whereas they are low when the substrates are metal based. There is no explanation for this observation. When the conducting substrates are changed in DSSC it makes significant changes in resistance of the cell, but no investigation on the resistance properties of the cell are found in this work.
The language used throughout the manuscript is very poor and not on international standard. The discussions are very poor and not informative. Therefore, the manuscript does not meet the quality criterion required for the journal and hence may be rejected.
Round 2
Reviewer 3 Report
Accept